# High-Fat-Diet-Induced Metabolic Disorders: An Original Cause for Neurovascular Uncoupling Through the Imbalance of Glutamatergic Pathways

**DOI:** 10.3390/biomedicines13071712

**Published:** 2025-07-14

**Authors:** Manon Haas, Maud Petrault, Patrick Gele, Thavarak Ouk, Vincent Berezowski, Olivier Petrault, Michèle Bastide

**Affiliations:** 1INSERM, CHU Lille, UMR-S1172-LilNCog-Lille Neuroscience & Cognition, University of Lille, F-59000 Lille, France; manon.haas.etu@univ-lille.fr (M.H.); maud.petrault@univ-lille.fr (M.P.); patrick.gele@univ-lille.fr (P.G.); thavarak.ouk@univ-lille.fr (T.O.); vincent.berezowski@univ-lille.fr (V.B.); olivier.petrault@univ-lille.fr (O.P.); 2UFR des Sciences, University of Artois, F-62300 Lens, France

**Keywords:** brain autoregulation, glutamate, metabolism, neurogliovascular coupling, nutrition

## Abstract

**Backgrounds/Objective:** The impact of metabolic disturbances induced by an unbalanced diet on cognitive decline in mid-life is now widely observed, although the mechanisms are not well identified. Here we report that glutamatergic vasoactive pathways are a key feature of high-fat-diet (HFD)-induced neurogliovascular uncoupling in mice. **Methods:** C57Bl6/J mice are fed either with normal diet (ND) or high-fat diet (HFD) during 6 or 12 months and characterized for metabolic status. Cerebral vascular tree from pial to intraparenchymal arteries, is investigated with Halpern’s arteriography and with differential interference contrast infrared imaging of brain slices. **Results:** A 70% alteration in the myogenic tone of the basilar artery is observed as early as 6 months (M6) after the HFD. Infrared imaging revealed a 77% reduction in the glutamate-induced vasodilation of intraparenchymal arterioles appearing after 12 months (M12) of the HFD. The respective contributions of enzymes involved in glutamatergic pathways were altered as a function of HFD and time. The decrease in astrocytic COX I observed at M6 was followed by a loss of neuronal COX II and a compensatory action of NOS at M12. **Conclusions:** This HFD-induced neurogliovascular uncoupling pathway offers therapeutic targets to consider for improving cerebral vasoactive functions while preventing peripheral metabolic disturbances.

## 1. Introduction

Cognitive health is one of the main concerns of the ageing population. It includes the well-being of memory, learning, and executive functions. To maintain cognitive health, the following behaviours are recommended: regular cognitive and physical activity throughout life, not smoking or drinking alcohol, and a healthy diet [1]. Indeed, it is now clear that a high-calorie diet is closely linked to weight gain and obesity, which affects one in eight people worldwide and is associated with the onset of several metabolic diseases [2]. Epidemiological studies have shown that people on high-fat diets develop cognitive problems from mid-life onwards, in association with metabolic disorders including weight gain [3,4,5,6]. It is therefore essential to understand the early causes of cognitive decline in this situation, before the onset of cerebrovascular lesions such as leakage of the blood–brain barrier (BBB), stroke, and neurodegeneration.

Since cerebral vessels are at the interface between peripheral metabolic disorders and the central nervous system, their alteration should occur before brain damage. In a previous longitudinal study, we showed that male C57Bl6/J mice fed a high-fat diet (HFD) until mid-life developed cognitive decline (mild impairment of visual recognition memory and memory flexibility), dysfunction of cerebral vasodilation with a simultaneous increase in visceral adipose tissue deposition, and weight gain [7]. In pathological situations such as obesity, peripheral disorders are well identified and can be directly linked to vascular dysfunction [8,9], but this is less the case for cerebral vessels. In our model, there were no signs of BBB rupture or reduction in cerebral perfusion, but there was an alteration in endothelium-dependent relaxation of the pial artery and intraparenchymal arterioles. In addition, some animal studies have demonstrated other mechanisms, such as a reduction in cerebral perfusion or in the diameter of cerebral arteries in models of rats or mice fed a high-fat diet [10,11].

An essential target would be neurogliovascular coupling (NGV), formed by neurons, interneurons, astrocytes, smooth muscle cells, and endothelial cells that link neuronal activities and cerebral blood flow (CBF). Neurogliovascular coupling is a vital ‘feed forward’ control mechanism involving neuronal signalling via neurotransmitters, which adjusts local CBF to the energetic needs of activated neurons. Any alteration in these mechanisms could lead to neurogliovascular uncoupling and subsequent cognitive decline. In this context, there is a clear interest in exploring the glutamate pathway as it is a key element in neurogliovascular coupling [12], targeting neurons, vessels, and glia. Glutamatergic pathways are therefore good indicators of coupling function. Glutamate is a potent vasodilator [13] through the following two different enzymatic pathways: (i) an astrocytic pathway, mediated by cyclooxygenase I (COX I) [14,15]; (ii) a neuronal pathway, involving nitric oxide synthase (NOS) [16] and cyclooxygenase II (COX II) [17,18]. Age-related uncoupling of NGV, involving the NO pathway, participates in age-related cognitive decline. There is growing interest in exploring potential alterations in different glutamate pathways in a metabolically disrupted environment.

The aim of the present study was to determine whether progressive metabolic disturbances induced by a long-term, high-fat diet result in glutamatergic neurogliovascular uncoupling. The metabolic profile of our mouse population (n = 136) was assessed to characterise the disturbances induced by the high-fat diet. After 6 (M6) or 12 (M12) months on the diet, we tested the vasoactive response to glutamate in the cerebral vascular tree from the pial arteries to the intraparenchymal arterioles. More specifically, different pharmacological modulators were used to assess the different enzymatic pathways acting at the glial, neuronal, and vascular levels. In parallel, the question of glutamate availability and neuroinflammation was explored.

## 2. Materials and Methods

### 2.1. Animal Feed and Experimental Design

All procedures were performed in accordance with the European Directive (2010/63UE), were approved by the local animal protection and use committee (Comité d’éthique en expérimentation animale des Hauts de France, Lille, France, reference: APAFIS#19866-201903071722644v2), and were in compliance with ARRIVE guidelines. Adult male C57Bl6/J mice (n = 144) aged 8 weeks at the start of the study (Janvier Labs, Le Genest Saint Isle, France), were divided into 2 diet groups during different sessions (4 sessions of 16 mice for M6 and 5 sessions of 16 mice for M12, 8 per diet group). A total of 3 mice died before six months on the diet (M6) and 5 others before twelve months on the diet (M12) for a total of 136 mice included in the study. The mice were housed in a standard cage under controlled temperature (21–24 °C) and humidity (40–70%) conditions, with a 12 h–12 h day/night cycle. At the start of the experiment, the mice were randomly assigned to a complete, high-fat diet (HFD) consisting of 40% (*w*/*w*) saturated fat, mainly from lard (HF231, Safe, Augy, France) or to a control diet (Normal Diet, ND, Safe, Augy, France) consisting of 3% (*w*/*w*) fat (A04, Safe, Augy, France) for the duration of the experiment (Table 1). Access to food and water was ad libitum. At M6 and M12, the mice were subjected to metabolic characterisation and spectrometry. They were then sacrificed for vascular analysis and immunohistochemistry (Figure 1).

### 2.2. Metabolic Profile

After 6 (M6) and 12 months (M12) of ND or HFD, the mice were subjected to the following tests to characterise their metabolic profile.

#### 2.2.1. Body Weight

From the start of the HFD or ND, the mice were weighed every 3 months until M6 or M12 depending on the protocol.

#### 2.2.2. Fasting Blood Glucose and Oral Glucose Tolerance Test (OGTT)

After 6 h of fasting [19], plasma glucose levels were determined using an Accu-Chek^®^ Performa glucometer system (Roche Diagnostic, Mannheim, Germany) on a blood sample taken from the tail vein of mice (T0). The mice were then orally administered 2 g/kg body weight of 0.1 g/mL D-glucose by gavage. Blood glucose levels were measured 10, 20, 30, 60, 90, and 120 min later. The area under the curve (AUC) was calculated. The accuracy of the system was checked before each session using an Accu-Check Guide Controls Glucose Calibrator (Roche Diagnostic, Mannheim, Germany) according to the manufacturer’s instructions.

#### 2.2.3. Plasma Metabolic Assays

The mice were fasted for 6 h. They were anaesthetised with 2% isoflurane to collect blood samples from the retro-orbital sinus. The blood samples were then centrifuged (1585× *g*, 20 min) and stored at −20 °C. Total cholesterol (CHOD-PAP 80106, Biolabo, Maizy, France) and triglycerides (GPO method 80019, Biolabo, Maizy, France) were measured using colorimetric enzyme assays from commercially available kits. A commercially available multicalibrator (Biolabo, Maizy, France) was used as a standard. Insulin levels were quantified by ELISA (Mercodia, Uppsala, Sweden) according to the standard curve.

#### 2.2.4. Hepatic Steatosis

After sacrifice by intraperitoneal (IP) injection of sodium pentobarbital (182.2 mg/kg), the livers were harvested and biopsies were taken. The livers were fixed for 24 h in 4% paraformaldehyde, dehydrated in 30% sucrose, and embedded in a matrix (OCT^®^). Biopsies were frozen in nitrogen liquid and subjected to frozen sectioning (8 µm, Leica CM3050S). Frozen sections were immersed in a 0.5% oil red O working solution for 15 min, counterstained with haematoxylin, then rinsed under tap water for 10 min. Photomicrographs were taken with a microscope (x20, DMLFS, Leica, Düsseldorf, Germany) from different views. The integrated optical density (IOD) of the oil red O-stained area was analysed using the free software ImageJ 1.54g.

#### 2.2.5. Visceral Adiposity

After sacrifice and removal of the liver, the peri-renal and peri-gonadal adipose tissues were removed and weighed. The sum of the two constitutes the visceral adipose tissue (VAT, g).

#### 2.2.6. Blood Pressure

Mean arterial pressure (MAP) was measured via the caudal artery in unanaesthetised animals (Blood Pressure Analysis System, Visitech Systems, Apex, NC, USA). Ten measurements were taken each day for three consecutive days to obtain an average.

### 2.3. Vasomotricity Studies

At M6 or M12 the mice were sacrificed by IP injection of sodium pentobarbital (182.2 mg/kg) and the brain was removed for cerebrovascular analysis.

#### 2.3.1. Ex Vivo Analysis of Basilar Artery Reactivity

##### Preparation

After anaesthesia, the brain was removed and placed in Krebs solution equilibrated with 20% O_2_/5% CO_2_/75% N_2_ (in mM): 119 NaCl, 24 NaHCO_3_, 4.7 KCl, 1.18 KH_2_PO_4_, 1.17 MgSO_4_, 7 H_2_O, 10 glucose, 1.6 CaCl_2_, and pH = 7.4. The basilar artery was dissected and ex vivo vasoreactivity was assessed using Halpern’s arteriograph (Living System Instrumentation, Burlington, VT, USA) on a proximal segment of the basilar artery (approximately 0.5–0.7 mm) [20]. The artery was attached to the proximal and distal cannulae with nylon ties and perfused with Krebs’ solution (37 °C). The distal cannula was closed to work under “no flow” conditions. The arteriography chamber was continuously supplied with Krebs solution. The proximal cannula was connected to a pressure transducer, a miniature peristaltic pump, and a servocontroller that continuously measured and adjusted transmural pressure. The entire arteriograph was placed on the stage of an inverted microscope equipped with a video camera and display screen. The diameter of the lumen was measured by dimensional analysis of the video image. The artery segment was stabilised for one hour at an intraluminal pressure of 20 mmHg.

##### Myogenic Tone

The myogenic tone of the basilar artery was assessed by increasing the intraluminal pressure from 25, 50, 75, and 100 mmHg and measuring the change in the intraluminal diameter of the artery (active diameter). At the end of the experiment, papaverine (10 µM, Pap, Sigma-Aldrich, Saint Quentin Fallavier, France), a phosphodiesterases inhibitor, was added to the medium before repeating the same pressure steps in order to induce a passive increase in intraluminal diameter (passive diameter). The myogenic tone is expressed as % constriction for each pressure step using the following equation [21]:(1)$$\%\;myogenic\;tone=\left(1-\frac{Active\;diameter}{Passive\;diameter}\right)\times 100$$

#### 2.3.2. Analysis of the Reactivity of Intraparenchymal Arterioles Ex Vivo

##### Preparation

After anaesthesia, the brain was removed and placed in a chamber containing an ice-cold solution of aCSF (in mM: 124 NaCl, 3.5 KCl, 2 MgSO_4_, 1.25 NaH_2_PO_4_, 2 CaCl_2_, 26 NaHCO_3_, 11 glucose, H_2_O 18 MΩ, and pH 7.4), which was continuously bubbled with carbogen (95 % O_2_/5 % CO_2_). Coronal slices (300 μm) were cut on a vibratome (World Precision Instruments, Stevenage, UK), then placed in a recovery chamber filled by aCSF at room temperature. The slices were continuously bubbled with carbogen and recovered for 60 min. Brain slices were transferred to a perfusion chamber (Warner Instruments, Harvard Apparatus, Les Ulis, France). Blood vessels were visualised using a Leica DMLFS microscope equipped with differential interference contrast (DIC) and an infrared-sensitive camera (Lumenera Infinity 3S-1URM, MicroMécanique SAS, Saint Germain en Laye, France). Selected vessels had at least one layer of vascular myocytes and a discernible luminal diameter of 10.42 ± 0.43 μm.

##### Pharmacological Modulations

Vessel precontraction was achieved using either 100 µM phenylephrine (Sigma-Aldrich) or a thromboxane A2 analogue U46619 2.5 µM (Tocris Biosciences, Noyal Châtillon sur Seiche, France). U46619 induces a more powerful and stable vasoconstriction than phenylephrine, closer to a response to a myogenic tone. The maximum vasodilatory effect of glutamate was induced using 100 µM (Sigma-Aldrich, Saint Quentin Fallavier, France). When maximal vasodilation was at a steady state, inhibitors of enzymes involved in neuronal and astrocytic glutamatergic pathways were used. In order to inhibit the astrocytic enzymatic component of vasodilation, 100 nM SC560 (Merck, Darmstadt, Germany), a selective COX I inhibitor, was infused over 4 min. Similarly, neuronal enzymatic components of vasodilation were assessed by NS398 25 µM (Merck), a selective COX II inhibitor, and Nω-Nitro-L-Arginine (L-NNA) 10 µM (Sigma-Aldrich), a NOS inhibitor. The first strategy assesses the respective contribution of these enzymes to glutamate-induced maximal vasodilation by applying the inhibitors cumulatively. The second strategy consists of isolated inhibition. Experiments were performed at 20–22 °C and the brain slice chamber was perfused at 7 mL/min.

The videos of the experiments were encoded to measure the diameter, offline images acquired in the same focal plane from 1 to 4 sectors were compared. The intraluminal diameters for each experimental condition were obtained by averaging the 2 measurements. In the different experiments, the number of arterioles studied corresponded to the number of slices used (only one arteriole per slice was examined).

### 2.4. Immunohistochemistry

For immunohistochemistry, mice of each food group and age had a different protocol (at M6: n_ND_ = 4 and n_HFD_ = 5; at M12: n_ND_ = 6 and n_HFD_ = 6). Mice were anaesthetised with sodium pentobarbital (182.2 mg/kg, IP) and perfused with heparinised saline. The brains were then harvested and post-fixed by immersion in 4% PFA (24 h, 4 °C) and dehydration in 30% sucrose (24 h, 4 °C). The brains were stored at −80 °C and subjected to frozen sections of the prefrontal cortex (20 µm thick coronal sections, Leica CM3050S, Nussloch, Germany). The prefrontal cortex sections were incubated overnight with a primary antibody of goat Glial Fibrillary Acidic Protein (GFAP, 1:1000, PA5-143587, InVitrogen, Les Ulis, France). The following day, they were incubated with a mouse anti-COX II primary antibody (1:50, Sc376861, Santa Cruz, Dallas, TX, USA) following the instructions in the Mouse on Mouse kit (M.O.M. kit, BMK-2202, Vector Laboratories, Newark, NJ, USA) for 12 h. Immunoreactions were visualised with the following secondary antibodies after incubation for 2 h: donkey anti-goat AlexaFluor 488 (A11055 Invitrogen) for GFAP revelation and rabbit anti-mouse AlexaFluor 568 (A11061 Invitrogen) for COX II revelation. Sections were mounted with Vectashield + DAPI (H-1200-10, Vector Laboratories). Brain sections were observed (x40) by confocal microscopy (Zeiss LSM 980 Airyscan2, Rueil Malmaison, France). Images were acquired using LSM Plus technology. Automated counting of the different cell types was performed using Fiji ImageJ (2.16.0) software by detecting the environment of the nuclei, followed by manual control.

### 2.5. Magnetic Resonance Spectrometry (MRS) Procedure

The experiments were performed on a 7.0 Tesla Animal Biospec MR scanner (Bruker, Ettlingen, Germany). In the MRI bed, isoflurane anaesthesia was administered using a face mask (isoflurane 1.5–2% and air 1.5 L/min). The animal’s vital parameters (oxygen saturation, pulse, rectal temperature, and respiratory rate) were monitored throughout the experiment. A gradient echo acquisition was first performed to confirm the positioning of the animal in the device. T2-weighted anatomical sequences were then performed in the axial and coronal planes to position the SRM acquisition voxel (2 × 2 × 1.5 mm^3^) in the anterior region of the brain (repetition time (TR)/echo time (TE) = 2500/33 ms, quadratic field of view = 4 cm, coded by a 256 × 256 quadratic matrix and 16 × 0.5 mm slices).

MRS acquisition and post-processing were performed using STEAM sequences (unsuppressed and suppressed STimulated Echo Acquisition Mode; TR/TE = 3500/3 ms, NEX = 512). Before each unsuppressed MR acquisition, a second-order shim was performed. Water suppression was performed using the VAPOR (VAriable Pulse power and Optimized Relaxation delays) technique and no external volume suppression module was used. The water peaks obtained with unsuppressed water acquisitions were used to check the spectral resolution and as an internal reference for estimating the quantity of metabolites. The signal-to-noise ratio (SNR) of the spectra for the ND vs. HFD groups were 45.39 ± 11.76 vs. 43.83 ± 10.88, respectively. The integrated area under the curve was used for quantification. The quality of the spectrum made it possible to evaluate the signals for the following metabolites: Glu (glutamate (1): 2.323 ppm and glutamate (2): 2.050 ppm), Gln (glutamine (1): 2.424 ppm and glutamine (2): 2.096 ppm), and common Glx (Glu + Gln: 3.750 ppm). MRS data were post-processed using JMRUI 5.2 software [22,23] while the AMARES algorithm (Advanced Method for Accurate, Robust, and Efficient Spectral fitting) was used to quantify the main metabolites [24].

### 2.6. Statistical Analysis

All values were expressed as mean ± S.E.M. Statistical analyses were performed using GraphPad Prism 10 (GraphPad Software Inc., La Jolla, CA, USA). Normality was assessed using the Shapiro–Wilk test. The effect of diet was assessed by two-way ANOVA and one-way ANOVA. Multiple comparisons were performed using the Šídák or Tukey multiple comparison tests. The threshold for statistical significance was set at *p* < 0.05.

### 2.7. Figures

The drawings presented in this document were created using BioRender (September 2024 release).

## 3. Results

### 3.1. HFD Induced Significant Metabolic Disturbances

There were no significant differences in body weight between the groups before administration of the experimental diets. During the experiment, weight gain was significantly higher in the HFD group than in the ND group from 3 months of the HFD (F_1, 537_ = 246.1, *p* < 0.001, Figure 2A). In HFD-fed mice, an increase in visceral adipose tissue deposition was observed at 6 and 12 months, with diet being a discriminating factor (F_1, 126_ = 144.6, *p* < 0.001, Figure 2B, Table 2). As shown in Figure 2C,D, oil red O staining revealed more pronounced steatosis in the livers of mice in the HFD group at both time points. Greater hepatic fat accumulation was quantified in this group (F_1, 111_ = 42.05, *p* < 0.001, Figure 2D, Table 2). Total blood cholesterol also increased significantly after 6 and 12 months of the HFD (F_1, 184_ = 50.66, *p* < 0.001, Table 2). In terms of carbohydrate metabolism, the HFD significantly increased fasting insulin levels (F_1, 182_ = 66.15, *p* < 0.001, Table 2) and fasting blood glucose levels (F_1, 202_ = 39.28, *p* < 0.001, Table 2). As shown in Figure 2E, the OGTT results showed that the HFD-fed mice had a delayed peak blood glucose level (25 min instead of 10 min in the ND group). This peak was higher than in the ND mice, and their blood glucose levels did not fall as quickly after 2 h, confirming the development of glucose intolerance. The area under the curve (AUC) was significantly increased by the HFD (F_1, 202_ = 47.17, *p* < 0.001, Figure 2F and Table 2). However, mean arterial pressure was not affected by the HFD at any time point (Table 2).

### 3.2. HFD Alters the Myogenic Tone in the Basilar Artery and Glutamate-Mediated Vasodilation in Parenchymal Arterioles

Pressure steps were imposed on the isolated basilar artery using Halpern’s arteriography after 6 or 12 months of the ND and HFD. The increase in pressure directly increased the intra-luminal diameter of the vessel (Figure 3A). In a normal (ND) configuration, myogenic tone developed rapidly, inducing smooth muscle cell constriction, as represented in the ND condition at M6 (left panel) and M12 (right panel) in Figure 3B. In contrast, in HFD-fed mice a significant decrease in the myogenic tone was observed as early as M6, in relation to diet (F_1, 48_ = 24.11, *p* < 0.001 at M6 and F_1, 51_ = 19.00, *p* < 0.001 at M12).

Using an ex vivo brain slice method, the maximal vasodilatory effect of glutamate was assessed on precontracted parenchymal arterioles from the prefrontal cortex region by applying 100 µM glutamate. The HFD induced a significant decrease in glutamate-induced dilation only at M12 (*p* < 0.05, Figure 3C). To assess the respective contribution of the main enzymes involved in glutamatergic dilation, three specific inhibitors were successively applied to brain slices (Figure 3D). COX I, COX II, and NOS were, respectively, inhibited by SC560 (100 nM), NS398 (25 µM), and L-NNA (10 µM) after the application of glutamate (100 µM). The charts represent the cumulative percentages of COX I, COX II, and NOS in the ND and HFD groups at M6 and M12. We observed a significant overall effect of the redistribution of the involvement of each enzyme pathway over time and diet (F_6, 72_ = 4.588, *p* < 0.001). Post hoc tests show that the COX-1 pathway is an independent factor that causes the observed changes in the redistribution of proportions of the other enzymatic pathways. Application of SC560 showed that the HFD significantly decreased the participation of COX I in the glutamate response at M6 (76.01% in ND-fed mice vs. 31.31% in HFD-fed mice, *p* < 0.01) and at M12 (57.05% in ND-fed mice vs. 29.13% in HFD-fed mice, *p* < 0.05). The contribution of COX II therefore tended to increase at M6 with HFD consumption (3.58% in ND mice to 18.44% in HFD mice) and with age in ND mice at M12 (18.54%). This trend became more pronounced in HFD-fed mice at M12 (29.12%). Similarly, the contribution of NO tended to increase with HFD consumption (at M6: 20.40% to 50.25% and at M12: 24.41% to 41.75%).

### 3.3. HFD Unbalances Glutamate Vasodilator Pathways

Isolated inhibition of key enzymes involved in maximal glutamate dilation was specifically performed to study astrocytic and neuronal pathways (Figure 4). Inhibition of COX I by SC560 (100 nM) confirmed the significant contribution (F_3,34_ = 5.862, *p* < 0.01, Figure 4A) of the astrocytic pathway in the glutamate-induced change in parenchymal arteriolar diameter in ND mice at M6 (−24.447 ± 7.290%) compared with the other groups (HFD_M6_: −7.449 ± 1.307%; ND_M12_: −11.413 ± 2.075%; HFD_M12_: −7.324 ± 1.797%). With regard to the neuronal component of glutamate-induced vasodilation, the use of L-NNA (10 µM) and NS398 (25 µM) highlighted the role of NO and COX II, respectively, in this vasodilation (Figure 4B,C). We confirmed a significantly greater involvement of NOS in mice fed the HFD at M6 (F_3, 52_ = 3.515, *p* < 0.05) compared with mice fed the ND at M6 (ND_M6_: −13.300 ± 1.402% vs. HFD_M6_: −7.817 ± 1.880%, *p* < 0.05) and at M12 (−7.515 ± 0.998%, *p* < 0.05). Similarly, the specific contribution of COX II was very low in ND-fed mice at M6 and became significantly higher with the HFD at M6 (F_3, 55_ = 3.599, *p* < 0.05). At M12, we observed a significant decrease in the contribution of COX II in HFD-fed mice (HFD = −10.380 ± 2.412% vs. ND = −3.699 ± 1.578%, *p* < 0.05).

### 3.4. HFD Induces Astrocytic Activation

Immunohistochemistry was used to observe the number of cells expressing astrocytic proteins such as GFAP (activated astrocytes) and COX II (Figure 5A,B). The number of COX II-positive cells increased significantly with diet (F_1, 317_ = 9.955, *p* < 0.01), more specifically at M6, where it doubled (ND_M6_: 8.3 ± 0.614 vs. HFD_M6_: 17.937 ± 2.026, *p* < 0.01). Interestingly, the number of COX II-positive cells increased significantly with time in the ND group (ND_M12_: 15.043 ± 1.321, *p* < 0.01). The number of GFAP-positive cells increased significantly with the HFD (F_1, 131_ = 6.073, *p* < 0.05), mainly at M6 (ND_M6_: 0.92 ± 0.341 vs. HFD_M6_: 3.0 ± 0.685, *p* < 0.05).

### 3.5. HFD Does Not Alter the Availability of Glutamate and Glutamine in the Brain

Quantitative analysis of the MRS spectra is shown in Figure 6A. Representative MRS spectra were obtained from a large 6 mm^3^ voxel located in the anterior region of the brain, comprising the prefrontal cortex and striatum (*Caude putamen*). Stable metabolites such as N-acetyl-aspartate (2.02 ppm) and total creatine (3.03 and 3.90 ppm) are commonly used as markers of neuronal activity and integrity. Quantification showed no difference between the ND and HFD groups whatever the duration of the diet (NAA: 290 10^−5^ ± 21 10^−5^ vs. 246 10^−5^ ± 34 10^−5^; total creatine: 199 ± 7 10^−5^ vs. 181 10^−5^ ± 3 10^−5^). The same glutamate levels were quantified in the anterior brain region independently of the diet groups (ND: 50 10^−5^ ± 6 10^−5^ vs. HFD: 60 10^−5^ ± 8 10^−5^, Figure 6B). No change in glutamine levels was observed in the ND (30 10^−5^ ± 5 10^−5^ vs. HFD: 30 10^−5^ ± 4 10^−5^, Figure 6B).

## 4. Discussion

This study shows for the first time dysfunctional myogenic tone in the basilar artery, as well as impaired glutamate vasodilation in intraparenchymal arterioles after 12 months of an HFD. These mice showed cognitive decline without any brain damage in our previous study [7] and the present results highlight the contribution of neurogliovascular uncoupling in diet-induced cognitive impairment. Glutamate vasodilator pathways were explored with specific inhibitions, and the results strongly suggest that HFDs and associated metabolic disturbances are responsible for an imbalance in these pathways. The vasomotor activity of the astrocytic enzyme COX I and the neuronal enzymes NOS and COX II is impaired, while glutamate availability is preserved. Overall, these results confirm that, in mid-life, poor dietary habits concomitant with weight gain can cause a glutamatergic neurogliovascular uncoupling.

The aim of the present study was to reproduce the ‘natural’ onset of metabolic disorders in mice given free access to a high-fat diet without physical exercise until mid-life. This objective was achieved as the HFD mice progressively gained more weight and developed more severe lipid and carbohydrate metabolic disorders than the ND mice. In a previous study involving 216 mice, we demonstrated that an HFD impairs the global vasoactive functions of pial arteries and cortical and hippocampal parenchymal arterioles [7]. Indeed, we had shown a significantly lower blood flow in the sylvian territory of HFD-fed mice (laser Doppler) after 6 and 12 months of an HFD, while no alteration in resting brain perfusion was observed (MRI). It is interesting to note that the reduction in blood flow was concomitant with the onset of metabolic disorders and cognitive decline, highlighting the link between peripheral disturbances and cerebral blood supply. This is highlighted here by the altered myogenic tone of the pial artery in HFD-fed mice from 6 months onwards, which worsened at 12 months. This result is confirmed by other studies, but in the mesenteric arteries specifically involving the vascular endothelium of the resistance arteries in the control of myogenic tone [25].

Since glutamate-induced vasodilation is significantly reduced by an HFD and the metabolic perturbations associated with M12 in our mice, we have demonstrated the occurrence of neurogliovascular uncoupling. The glutamate vasodilator mechanism is mediated by both an astrocytic pathway involving COX I and a neuronal pathway involving COX II and NOS [26]. Astrocytic activation by glutamate could be the result of its binding to metabotropic receptors (mGluR2/3) and/or astrocytic excitatory glutamate amino acid transporters 1 and 2 (EAAT1 and EAAT2). At the astrocytic level, the release of glutamate induces an increase in intracellular calcium, which activates COX I. This enzyme then releases various vasodilatory mediators, such as epoxyeicosatrienoic acids (EETs) and prostaglandins [14,15]. The link between glutamate release and elevated intracellular calcium is not well known [27]. Using a specific enzyme inhibitor, we demonstrated a drastic decrease in the contribution of COX I to glutamatergic vasodilation under an HFD. This decrease in astrocytic contribution was observed at M6 and maintained at M12, raising the question of a compensatory activation of the neuronal enzymatic pathway in residual vasodilation, including NOS and COX II.

Binding of glutamate to ionotropic receptors (NMDARs) on post-synaptic neurons allows calcium-dependent nNOS to release NO onto vascular smooth muscle cells, inducing vasodilation [16]. The increase in intracellular calcium following NMDAR activation can activate COX II, which then metabolises arachidonic acid (AA) to prostaglandin E2 (PGE2), inducing vasodilation via the activation of vascular EP2-4 receptors [17,18,28]. Here, we demonstrated that an HFD significantly increased the contribution of NOS in glutamate vasodilation as well as the contribution of COX II to M6. This enzymatic reorganisation provides primary compensation for the altered astrocytic pathway. This compensation appears to be limited since we observed a reduction in the potential activity of these enzyme systems at M12.

Another finding is the overexpression of astrocytic markers after an HFD. When highly expressed, astrocytic markers may be linked to neuroinflammation, a complex phenomenon initiated by the production of pro-inflammatory cytokines, but also by the oxidative stress of lipid metabolites [29,30]. We observed astrocytic hyperactivation and an increase in COX II expression in the prefrontal cortex at M6, which was maintained at M12. In our context, astrocytes are key intermediaries in neurogliovascular coupling and cerebral energy metabolism, and the hyperactivation observed is more consistent with metabolic overload [31]. Changes in lipid status involving saturated fatty acids are known to disrupt cerebral energy metabolism, which may have pathological consequences, as astrocytes are the metabolic suppliers of neurons, which could be affected by the reduced contribution of COX I to neurovascular coupling. In addition, the increased expression of COX II leads to an unbalanced production of vasoactive metabolites and could probably also contribute to the observed neurogliovascular uncoupling. Consistent with this, the compensatory contribution of NO to glutamate-mediated vasodilation, established at M6, is lost at M12. NO is highly sensitive to the redox state of cells. Oxidative stress could therefore explain our result, but this hypothesis requires further investigation.

## 5. Conclusions

Taken together, our results highlight the existence of neurovascular changes following long-term high-fat feeding, which could be responsible for the lesion-free cognitive decline observed in our middle-aged mice. The neurovascular mechanisms involved, including an altered astrocytic glutamatergic pathway followed by a loss of the compensatory action of neuronal NOS, represent potential therapeutic targets to be considered alongside preventive care for peripheral metabolic disorders. Patients with multiple metabolic disorders in mid-life should be screened for neurogliovascular uncoupling to help prevent pathological progression of their cognitive health.

## Figures and Tables

**Figure 1 biomedicines-13-01712-f001:**
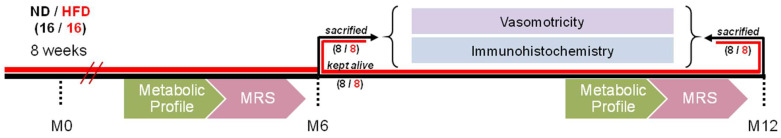
Overview of the experimental design. This schematic representation shows 2 sessions of 16 mice divided into 2 diet groups. The first session stops at 6 months of the diet, the second at 12 months of the diet.

**Figure 2 biomedicines-13-01712-f002:**
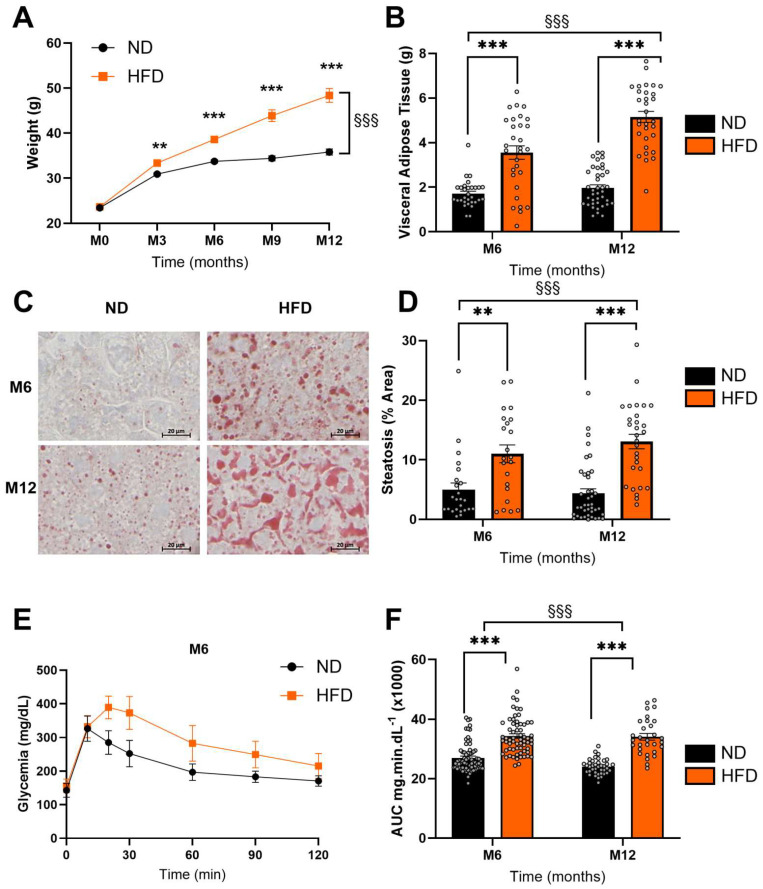
HFD induces a disturbance in lipid and glucose homeostasis. (**A**) Body weight gain (g) over 12 months as a function of time in male C57Bl6/J mice fed a normal diet (ND) or a high-fat diet (HFD). (**B**) Visceral adipose tissue accumulation (VAT, in g) after 6 and 12 months of ND (n_ND_ = 69) or HFD (n_HFD_ = 61). (**C**) Lipid infiltration in the liver of ND and HFD mice, stained with oil red O (x20). Representative illustrations of liver sections from ND and HFD mice after 6 months of diet. (**D**) Quantification of lipid accumulation in the livers of ND (n_ND_ = 64) and HFD (n_HFD_ = 51) mice at M6 and M12. (**E**,**F**) Oral glucose tolerance test. (**E**) Representative kinetics of blood glucose values obtained in ND and HFD mice shown at M6. (**F**) Quantification of area under the curve (AUC) (mg min dL^(−1)^ of ND (n = 64) and HFD (n = 56) mice at M6 and M12. Data are presented as mean ± S.E.M. Statistical analysis was performed using a two-way ANOVA, §§§ *p* < 0.001 for the effect of diet factor. Student’s *t*-test was used to compare the weight of mice at each time point, *** *p* < 0.001. Šídák’s multiple comparisons test was performed for the HFD vs. ND comparison, ** *p* < 0.01, *** *p* < 0.001.

**Figure 3 biomedicines-13-01712-f003:**
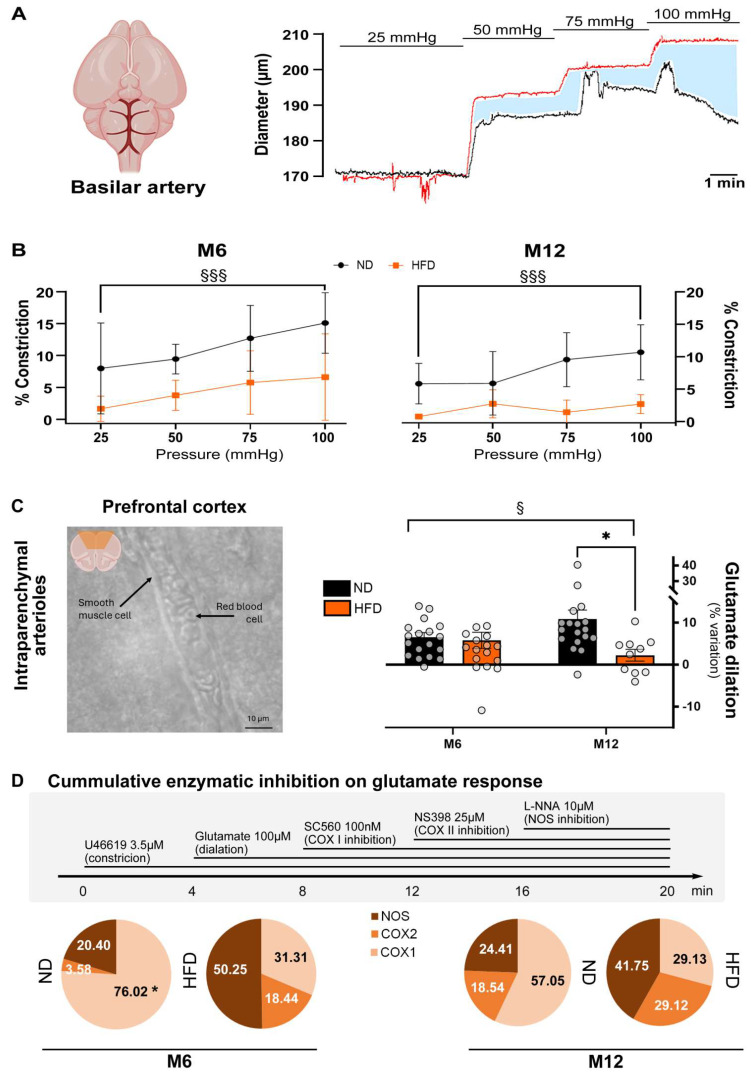
The HFD alters the myogenic tone of the basilar artery and the maximal vasodilatory effect of glutamate. (**A**). Representative traces of the basilar artery diameter in response to increasing pressure steps in the absence (black line) or presence (red line) of papaverine to assess the myogenic tone. (**B**). Basilar artery myogenic tone is shown at M6 (left panel) and M12 (right panel) for mice in the normal (ND) or high-fat diet (HFD) groups. (**C**). The maximal effect of glutamate was measured using an ex vivo brain slice model in ND and HFD mice at M6 (n_ND_ = 19, n_HFD_ = 19) and M12 (n_ND_ = 19, n_HFD_ = 11). (**D**). Percentage contributions of vasoactive enzymes involved in the glutamate response. COX I, COX II, and NOS were successively inhibited by SC560 (100 nM), NS398 (25 µM), and L-NNA (10 µM), respectively, after application of glutamate (100 µM). The charts represent the cumulative percentages of COX I, COX II, and NOS in the ND and HFD groups at M6 and M12. Data are presented as mean ± S.E.M. Statistical analysis was performed using a two-way ANOVA (for myogenic tone and for maximal glutamate response) §§§ *p* < 0.001 and § *p* < 0.05 for the effect of the diet factor, and two-way ANOVA (for the percentage contributions of vasoactive enzymes) $$$ *p* < 0.01 and post hoc ** p* < 0.05 for the contribution of COX I (ND vs. HFD).

**Figure 4 biomedicines-13-01712-f004:**
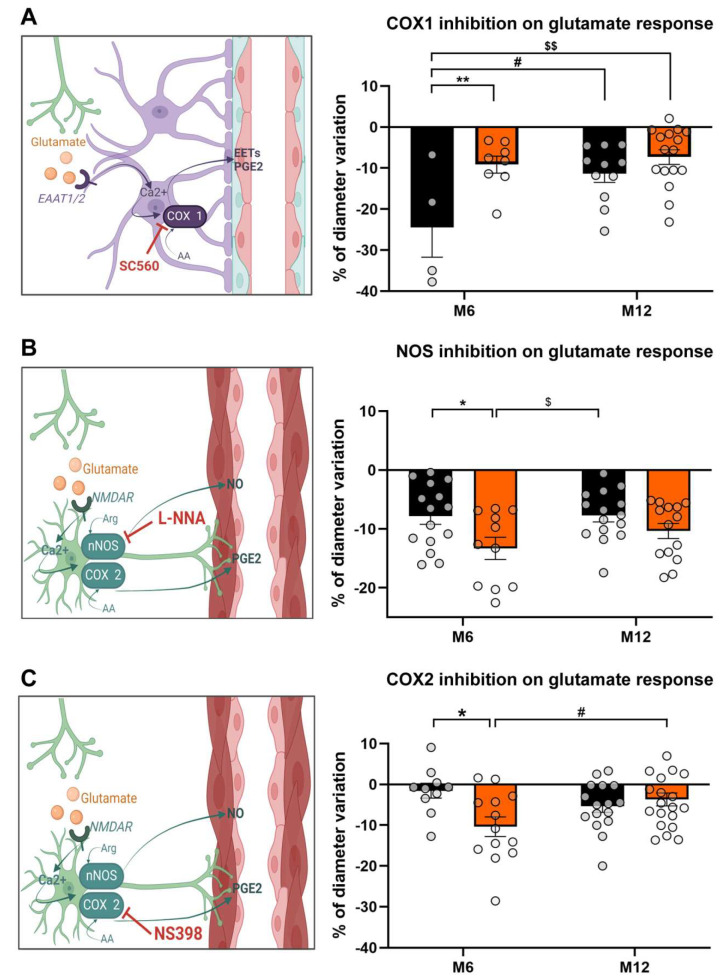
HFD induces changes in glutamatergic pathways. Specific inhibition of vasoactive enzymes involved in the glutamate response. (**A**) Probable astrocytic pathway involving COX I (**left**), specific contribution of COX I assessed by SC560 100 nM in glutamate-induced diameter change (right panel n_ND_ = 4; n_HFD_ = 7 at M6 and n_ND_ = 11; n_HFD_ = 16 at M12). (**B**) Probable neuronal pathway involving NOS (**left**), specific contribution of NOS assessed by 10 µM L-NNA in glutamate-induced diameter change (right panel n_ND_ = 10; n_HFD_ = 13 at M6 and n_ND_ = 16; n_HFD_ = 20 at M12). (**C**) Probable neuronal pathway involving COX II (**left**), specific contribution of COX II assessed by NS398 25 µM in glutamate-induced diameter change (right panel n_ND_ = 15; n_HFD_ = 11 at M6 and n_ND_ = 15; n_HFD_ = 14 at M12). Data are presented as mean ± S.E.M for ND and HFD mice at M6 and M12. Statistical analysis was performed using one-way ANOVA, ** *p* < 0.01, * *p* < 0.05 for ND versus HFD at M6. # *p* < 0.05 for M6 versus M12. $ *p* < 0.05 for HFD at M6 versus ND M12, $$ *p* < 0.01 for ND M6 versus HFD M12.

**Figure 5 biomedicines-13-01712-f005:**
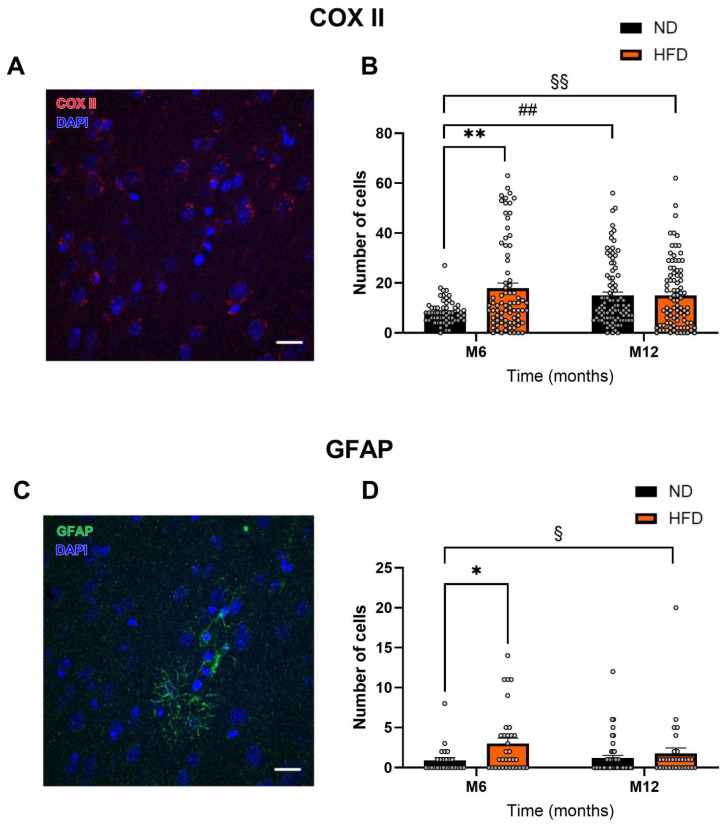
HFD and obesity increase the expression of COX II and GFAP. (**A**,**C**) Representative illustrations of immunohistochemistry tests performed on COX II (**A**) and GFAP (**C**). (**B**,**D**) Number of cells expressing COX II (**B**) and GFAP (**D**) in the ND and HFD groups at M6 and M12. Data are presented as mean ± S.E.M. Statistical analysis was performed using a two-way ANOVA, §§ *p* < 0.01, § *p* < 0.05, for the effect of the “diet” factor. Šídák’s multiple comparison test was performed: ** *p* < 0.01, * *p* < 0.05 for HFD vs. ND and ## *p* < 0.01 for M6 vs. M12.

**Figure 6 biomedicines-13-01712-f006:**
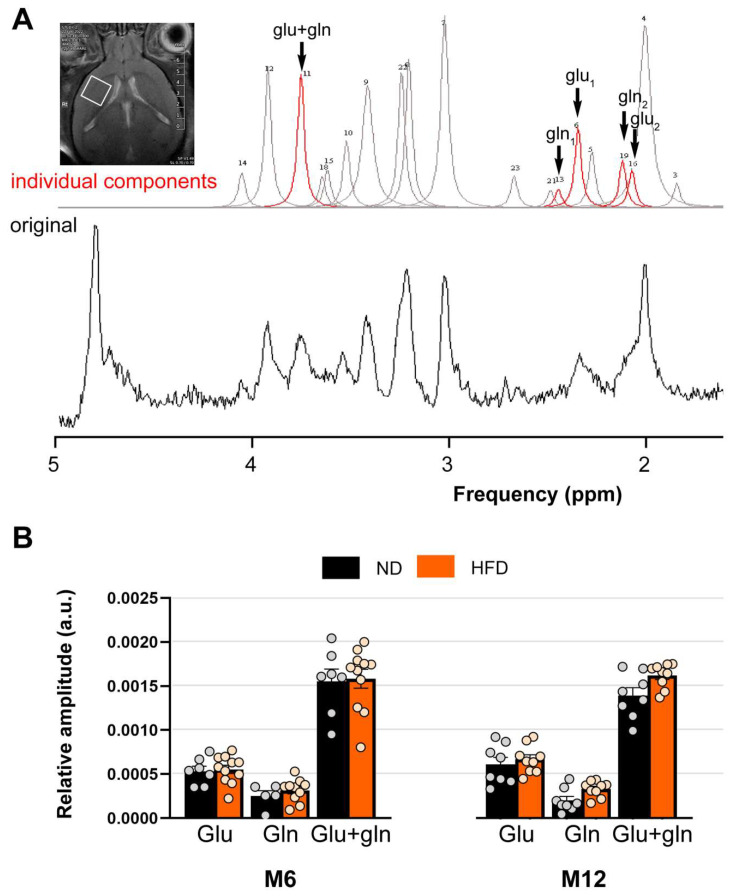
HFD does not alter the availability of glutamate and glutamine in the brain. (**A**). Estimated (upper trace) and original (lower trace) representative traces extracted from the quantitative analysis of MRS spectra performed from a large 6 mm^3^ voxel located in the anterior region of the brain. (**B**). Relative magnitudes (in arbitrary units) of glutamate and glutamine levels in the anterior brain region for the normal diet (n_ND_ = 7) and the high-fat diet (n_(HFD)_) = 11) at M6 and the normal diet (n_ND_ = 8) and the high-fat diet (n_HFD_ = 9) at M12. Data are presented as mean ± S.E.M. for ND and HFD mice at M6 and M12. Statistical analysis was performed using one-way ANOVA and no significant differences were found between groups.

**Table 1 biomedicines-13-01712-t001:** Nutritional and energy profiles of the normal diet (ND) and the high-fat diet (HFD).

	Proteins	Lipids	Carbohydrates	Minerals	Cellulose	Humidity	Total
**ND**							
**Nutrients (%)**	16.2	3.2	60.5	4.6	3.9	11.6	100
**Energy (%)**	19.3	8.4	72.3				100
**Energy (kcal/kg)**	648	288	2420				3326
**HFD**							
**Nutrients (%)**	28.7	40.1	14.2	8.2	3.1	5.6	100
**Energy (%)**	21.3	70.7	8				100
**Energy (kcal/kg)**	11.48	3620.2	567.6				5335.8

**Table 2 biomedicines-13-01712-t002:** Metabolic characterisation of ND and HFD mice at M6 and M12.

M12	M6		Time vs. Diet
73.71 ± 1.89 (23)	77.11 ± 1.41 (48)	ND	Mean Arterial Pressure (mmHg)
64.95 ± 1.58 (23)	77.14 ± 1.51 (40)	HFD	
1.98 ± 0.14 (39)	1.71 ± 0.11 (30)	ND	Visceral Adipose Tissue (g) §§§
5.18 ± 0.25 (31) ***	3.56 ± 0.30 (30) ***	HFD	
4.364 ± 0.785 (40)	5.01 ± 1.114 (24)	ND	Liver lipid accumulation (% Area) §§§
13.08 ± 1.22 (29) ***	11.01 ± 1.51 (22) **	HFD	
65.41 ± 5.84 (39)	65.11 ± 3.28 (64)	ND	Triglycerides (mg.dL^−1^) §§
49.61 ± 2.82 (31) *	57.09 ± 2.86 (56)	HFD	
148.0 ± 6.17 (39)	151.0 ± 5.123 (64)	ND	Total cholesterol (mg.dL^−1^) §§§
260.8 ± 8.41 (31) ***	245.9 ± 7.45 (56) ***	HFD	
1.293 ± 0.15 (39)	1.194 ± 0.10 (64)	ND	Insulin (µg.L^−1^) §§§
3.807 ± 0.50 (27) ***	2.116 ± 0.17 (56) ***	HFD	
140.7 ± 2.51 (39)	144.1 ± 2.89 (72)	ND	Fasting glycemia (mg.dL^−1^) §§§
170.2 ± 4.44 (31) ***	158.5 ± 3.27 (64) ***	HFD	
24.027 ± 0.41 (39)	27.03 ± 0.58 (71)	ND	OGTT (AUC. mg.min.dL^−1^) §§§
34.092 ± 1.11 (30) **	35.33 ± 0.80 (64) ***	HFD	

Results are expressed in mean ± IC_95_. Statistical analysis was performed using the Mann–Whitney test, §§§ *p* < 0.001, §§ *p* < 0.01 for the effect of the diet factor. Multiple Mann–Whitney tests were performed: * *p* < 0.05, ** *p* < 0.01, *** *p* < 0.001 for HFD vs. ND comparison. The numbers of mice involved in each measure are indicated in brackets.

## Data Availability

The original contributions presented in this study are included in the article. Further inquiries can be directed to the corresponding author.

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
