# Peer review of "High-Fat-Diet-Induced Metabolic Disorders: An Original Cause for Neurovascular Uncoupling Through the Imbalance of Glutamatergic Pathways"

_biomedicines, 2025, doi:10.3390/biomedicines13071712_

Round 1

Reviewer 1 Report

Comments and Suggestions for Authors

The manuscript is very interesting and provides a specific mechanism that connects metabolic disorders with cognitive decline. I have just a few suggestions for the authors:

  1. Introduction: line 38: The phrase “an enriched diet is closely linked to weight gain and obesity” could be misleading. “Enriched diet” can also refer to foods with added nutrients or health-promoting components. The authors may consider specifying what the diet is enriched with (e.g., fat, sugar, or calories), as this would help clarify the intended meaning and reduce ambiguity.
  2. Table 1: Please replace the commas with dots to indicate decimal points, in accordance with standard English-language scientific formatting.
  3. In Figure 2 caption, it is written * p < 0.05 for COX I contribution (ND 320 M6 vs. HFD M12). It is somewhat unclear why this specific significant difference is singled out, as other significant comparisons are mentioned in the main text. The authors may consider revising the caption to ensure consistency with the results described, or briefly clarifying why this particular difference is emphasized.
  4. The font sizes in the figure captions are inconsistent; please standardize them.
Comments on the Quality of English Language

There are some grammar issues, such as:

    1. Line 66-67: “Neurogliovascular coupling is a vital feed forward control mechanism involving neuronal signalling via neurotransmitters which adjusts local CBF to the energy requirements of activated neurons.”: “feed forward” should be hyphenated (“feed-forward”), and a comma is needed before “which”.
    2. Lines 470–471: The sentence “In line with this, the compensatory NO contribution to glutamate vasodilation was installed at M6 is lost at M12” appears to contain a grammatical issue that may hinder clarity. The authors may consider rephrasing it for improved readability. For example: “In line with this, the compensatory NO contribution to glutamate-induced vasodilation that was established at M6 is lost by M12.”

In light of these examples, I suggest a careful read-through to catch and correct any remaining minor issues related to grammar, punctuation, or clarity that I may not have identified.

Author Response

We will thank you for your comments. We have taken them into account and modified the text as suggested. All the changes or modifications of the text have been underlined in green. 

The manuscript is very interesting and provides a specific mechanism that connects metabolic disorders with cognitive decline. I have just a few suggestions for the authors:

Comment 1 : Introduction: line 38: The phrase “an enriched diet is closely linked to weight gain and obesity” could be misleading. “Enriched diet” can also refer to foods with added nutrients or health-promoting components. The authors may consider specifying what the diet is enriched with (e.g., fat, sugar, or calories), as this would help clarify the intended meaning and reduce ambiguity.

Response 1 : The sentence has been modified as suggested and, by extension, the whole paragraph has been revised to remove and/or reduce any ambiguity.

Comment 2 : Table 1: Please replace the commas with dots to indicate decimal points, in accordance with standard English-language scientific formatting.

Response 2 : It has been modified throughout the text.

Comment 3 : In Figure 2 caption, it is written * p < 0.05 for COX I contribution (ND 320 M6 vs. HFD M12). It is somewhat unclear why this specific significant difference is singled out, as other significant comparisons are mentioned in the main text. The authors may consider revising the caption to ensure consistency with the results described, or briefly clarifying why this particular difference is emphasized.

Response 3 : Statistic representations, values in the main text, figure 2 and its legend have been clarified.

Comment 4 : The font sizes in the figure captions are inconsistent; please standardize them.

Response 4 : The font sizes have been harmonized.

Comment 5a : Comments on the Quality of English Language. There are some grammar issues, such as:Line 66-67: “Neurogliovascular coupling is a vital feed forward control mechanism involving neuronal signalling via neurotransmitters which adjusts local CBF to the energy requirements of activated neurons.”: “feed forward” should be hyphenated (“feed-forward”), and a comma is needed before “which”.

Response 5a : The sentence has been modified as suggested.

Comment 5b: Lines 470–471: The sentence “In line with this, the compensatory NO contribution to glutamate vasodilation was installed at M6 is lost at M12” appears to contain a grammatical issue that may hinder clarity. The authors may consider rephrasing it for improved readability. For example: “In line with this, the compensatory NO contribution to glutamate-induced vasodilation that was established at M6 is lost by M12.”

Response 5b : The sentence has been modified as suggested.

Comment 6: In light of these examples, I suggest a careful read-through to catch and correct any remaining minor issues related to grammar, punctuation, or clarity that I may not have identified.

Response 6: The entire manuscript has been carefully proofread to improve the quality of the English.

Reviewer 2 Report

Comments and Suggestions for Authors

The manuscript is interesting but some modifications are recommended.

  1. In abstract "The influence of diet-induced metabolic disorders on cognitive decline at midlife is now widely observed, but the mechanisms at play are still not identified". The sentence is grammarly weak and confusing. please paraphrase it.
  2. In line 37 " (“Cognitive Health and Older Adults,” 2020), is that a reference? if so, it should be written as other format. If not, please remove it.
  3. The introduction contains too much information about the authors' previous work. I suggest summarizing this information and focus more on the current study.
  4. At the end of the introduction, you should state the aim of the work in a concise and clear form rather than writing about the methodology. Therefore, write the aims clearly in a separate paragraph and no need to mention any methodology.
  5. In Animal Diet and Cohorts, n=136 adult male C57Bl6/J mice, reported in different cohorts (n= 9 cohorts). You must clarify how many groups or cohort were there? if it equally divided, each group will be 15.111, which is not logic, and if they were not equally distributed, it is not logic and why?
  6. Line 98, were housed, this must be a new start and changed to "The mice were housed....
  7. clarify what is the exact composition of high-fat diet.
  8. The authors should add experimental design overview or construct a chart to clarify how many groups, the diet of each group, and the overall time of the experiment. Again i found it confusing to use 9 mice per cohort and you mentioned only two diets: HFD and ND.
  9. In Metabolic Profile line 105, the authors must clearly state what are exactly measured and which kits and instruments were used to ensure reproducibility.
  10. In body weight, it is confusing again not to know the overall time of the experiment; every 3 months out of how much?
  11. In fasting glycemia, it is well known that fasting blood glucose measured typically after 8-12 hours of fasting. Why you detect it after only 6 hours?
  12. The subheading "Plasmatic Metabolic Dosages" is not appropriate. There is no dosage?!!
  13. What is your reference for the used equation to calculate %myogenic tone?
  14. Why you expressed your results as SEM and not as mean ± SD.
  15. P value of significance should be italic throughout the manuscript.
  16. Panel C in Fig. 1 should be of higher resolution.
  17. Uniform the font and its size in all figure legends.
  18. Write the conclusion in a separate heading after the discussion.
  19. English language and grammars should be revised throughout the manuscript.
Comments on the Quality of English Language
  1. English language and grammars should be revised throughout the manuscript.

Author Response

Answers to Reviewer 2

We will thank you for your comments. We have taken them into account. We have modified the text as suggested and added a figure to explain the experimental design and a table for the details of the composition of the two diets. All the changes or modifications of the text have been underlined in green.

The manuscript is interesting but some modifications are recommended.

Comment 1 : In abstract "The influence of diet-induced metabolic disorders on cognitive decline at midlife is now widely observed, but the mechanisms at play are still not identified". The sentence is grammarly weak and confusing. please paraphrase it.

Response 1 : We have rephrased the sentence.

Comment 2 : In line 37 " (“Cognitive Health and Older Adults,” 2020), is that a reference? if so, it should be written as other format. If not, please remove it.

Response 2 : NIH (2024) and WHO (2022) NIH are websites that have been listed in the bibliography with the URL and date of consultation.

Comment 3 : The introduction contains too much information about the authors' previous work. I suggest summarizing this information and focus more on the current study.

Response 3 : We have reworded the paragraph in the introduction referring to our previous work so as to summarise the main findings that are essential for an understanding of this study.

Comment 4 : At the end of the introduction, you should state the aim of the work in a concise and clear form rather than writing about the methodology. Therefore, write the aims clearly in a separate paragraph and no need to mention any methodology.

Response 4 : In order to be clearer about the aim of our work, we have separated and rewritten the last paragraph, focusing on the study strategy and removing the methodological aspects.

Comment 5 : In Animal Diet and Cohorts, n=136 adult male C57Bl6/J mice, reported in different cohorts (n= 9 cohorts). You must clarify how many groups or cohort were there? if it equally divided, each group will be 15.111, which is not logic, and if they were not equally distributed, it is not logic and why?

Response 5 : The distribution of mice has been clarified in « Animal diet and cohorts ». Nine sessions of 16 mice (8 mice fed with a Normal Diet and 8 mice fed with a High Fat Diet) have been used in this study. Mice were followed for either 6 or 12 months

Comment 6 : Line 98, were housed, this must be a new start and changed to "The mice were housed....

Response 6 : The sentence has been modified.

Comment 7 : Clarify what is the exact composition of high-fat diet.

Response 7 : We have added a table (Table 1) describing the composition of the two diets, the normal one (ND) and the high fat one (HFD) diets with the energy contribution of the different components.

Comment 8 : The authors should add experimental design overview or construct a chart to clarify how many groups, the diet of each group, and the overall time of the experiment. Again I found it confusing to use 9 mice per cohort and you mentioned only two diets: HFD and ND.

Response 8 : A new figure (Figure 1) has been added with a design protocol of the study highlighting the two diet groups (ND and HFD), the two time-points of experiments (6 and 12 monts) with the detail of the performed experiments : Metabolic profile characterisation ; Magnetic resonance spectrometry ; Vasomotricity ; Immunohistochemistry.

Comment 9 : In Metabolic Profile line 105, the authors must clearly state what are exactly measured and which kits and instruments were used to ensure reproducibility.

Response 9 : Each parameter was described in a sub-paragraph. The origin of the different enzymatic kits and ELISA test was specified. The kits or ELISA test always come from the same suppliers. For each of them, the standard calibrators given by the supplier (cholesterol and triglycerides kits) were used to ensure the reproducibility of the results. For the insulinemia, a standard curve is performed at each assay.

Comment 10 : In body weight, it is confusing again not to know the overall time of the experiment; every 3 months out of how much?

Response 10 : The mice were weighted every 3 months from Day0 to 6 or 12 months. We have modified the sentence.

Comment 11 : In fasting glycemia, it is well known that fasting blood glucose measured typically after 8-12 hours of fasting. Why you detect it after only 6 hours?

Response 11 : The choice of this 6h fasting duration was based on the article cited here (Andrikipoulos et al Evaluating the glucose tolerance test in mice (2008) Am J Physiol Endocrinol Metab 295 : E1323-E1332). In this study, the authors evaluated different fasting durations to determine the optimal conditions for assessing glucose tolerance in C57Bl6 mice fed with normal or HF diet. A fasting duration of 6 h appears to be optimal for the OGTT test in mice.

Comment 12 : The subheading "Plasmatic Metabolic Dosages" is not appropriate. There is no dosage?!!

Response 12 : Total cholesterol, triglycerides and insulin have been quantified in the plasma of the mice either by colorimetric method (cholesterol and triglycerides) or by ELISA assay (insulin).

Comment 13 : What is your reference for the used equation to calculate %myogenic tone?

Response 13 : The equation used to calculate the % myogenic tone comes from the following publication, which we have added to the bibliography of the article «  Reperfusion decreases myogenic reactivity and alters middle cerebral artery function after focal cerebral ischemia in rats » (2006) Cipolla et al, Stroke 28(1) : 176-180.

Comment 14 : Why you expressed your results as SEM and not as mean ± SD.

Response 14 : All bars were illustrated with individual dots giving obvious descriptive informations about population distribution and also the standard deviation (mean(SD)). We have chosen SEM to qualify the statistical precision of the estimated mean.

Comment 15 : P value of significance should be italic throughout the manuscript.

Response 15 : All the P value significance are in italics.

Comment 16 : Panel C in Fig. 1 should be of higher resolution.

Response 16 : The quality of the liver images has been improved in Figure 2.

Comment 17 : Uniform the font and its size in all figure legends.

Response 17 : The font and the size have been standardized throughout the figures.

Comment 18 : Write the conclusion in a separate heading after the discussion.

Response 18 : A separate paragraph with the heading « Conclusion » have been added after the discussion part.

Comment 19 : English language and grammars should be revised throughout the manuscript.

Response 19 : The entire manuscript has been carefully proofread to improve the quality of the English.

Round 2

Reviewer 2 Report

Comments and Suggestions for Authors

Thank you for your effort to address the comments, but still some comments not clearly responded.

  1. your response for comment # 5 "The distribution of mice has been clarified in « Animal diet and cohorts ». Nine sessions of 16 mice (8 mice fed with a Normal Diet and 8 mice fed with a High Fat Diet) have been used in this study. Mice were followed for either 6 or 12 months".... It is still not clear why the total number of the mice were 136 and in your response, you said 9 sessions of 16 mice, if so, they should be 144?! Moreover, what do you mean by 9 session if you also responded that the mice were followed for either 6 or 12 months???.... The authors must make this point clearer.
  2. Regarding comment 12, 

    Comment 12 : The subheading "Plasmatic Metabolic Dosages" is not appropriate. There is no dosage?!!

    Response 12 : Total cholesterol, triglycerides and insulin have been quantified in the plasma of the mice either by colorimetric method (cholesterol and triglycerides) or by ELISA assay (insulin).
    the response is not related to the comment, I did not ask what did you determine or how. The authors just have to change the subheading title.

Author Response

Comment 1 : your response for comment # 5 "The distribution of mice has been clarified in « Animal diet and cohorts ». Nine sessions of 16 mice (8 mice fed with a Normal Diet and 8 mice fed with a High Fat Diet) have been used in this study. Mice were followed for either 6 or 12 months".... It is still not clear why the total number of the mice were 136 and in your response, you said 9 sessions of 16 mice, if so, they should be 144?! Moreover, what do you mean by 9 session if you also responded that the mice were followed for either 6 or 12 months???.... The authors must make this point clearer.

Response 1 : To clarify the total number of animals included in the study, as well as their distribution across the different sessions and times of interest, we have added the following sentence highlighted in blue to the main text:

"Adult male C57Bl6/J mice (n= 144) aged 8 weeks at the start of the study (Janvier Labs, Le Genest Saint Isle, France), were divided into 2 diet groups during different sessions (4 sessions of 16 mice for M6 and 5 sessions of 16 mice for M12, 8 per diet group). Three mice died before six months on the diet (M6) and 5 others before twelve months on the diet (M12) for a total of 136 mice included in the study.”

Comment 2 : Regarding comment 12, The subheading "Plasmatic Metabolic Dosages" is not appropriate. There is no dosage?!! The response is not related to the comment, I did not ask what did you determine or how. The authors just have to change the subheading title.

Response 2 : As request we changed the subeading title : 2.2.3. Plasma metabolic assays